# Zn (II) Porphyrin Built-in D–A Covalent Organic Framework for Efficient Photocatalytic H_2_ Evolution

**DOI:** 10.3390/polym14224893

**Published:** 2022-11-13

**Authors:** Mingbo Lv, Xitong Ren, Ronghui Cao, Zhiming Chang, Xiao Chang, Feng Bai, Yusen Li

**Affiliations:** 1Key Laboratory for Special Functional Materials of Ministry of Education, National & Local Joint Engineering Research Center for High-Efficiency Display and Lighting Technology, School of Materials Science and Engineering, Collaborative Innovation Center of Nano Functional Materials and Applications, Henan University, Kaifeng 475004, China; 2College of Chemistry and Chemical Engineering, Luoyang Normal University, Luoyang 471934, China

**Keywords:** covalent organic framework, porphyrin, photocatalytic hydrogen evolution, donor–acceptor, photocatalyst

## Abstract

Covalent organic frameworks (COFs) with donor–acceptor (D–A) units are credible photocatalysts for their per-designed structure, inherent porosity, large surface area, splendid stability and so forth. Developing COFs with an excellent photocatalytic efficiency for hydrogen evolution is of a great significance in alleviating the energy crisis. Herein, a D–A type imine-linked crystalline Zn-Por-TT COF was fabricated successfully via the co-polymerization of electron-deficient Zinc (II) 5,10,15,20-tetrakis(para-aminophenyl) porphyrin (Zn-TAPP), and electron-rich thieno[3,2-b]thiophene-2,5-dicarbaldehyde (TT). Profiting from the D–A complex structure, the obtained Zn-Por-TT COF showcases an excellent photocatalytic activity with a hydrogen evolution rate of 8200 μmol/g/h, while the Zn-TAPP monomer presents practically no capacity for the generation of hydrogen under identical conditions. In addition, the counterparts Por-TT COF and COF-366-Zn were employed to illustrate the enhancement of the photocatalytic performance by metal catalytic sites and D–A structures. In addition, the counterparts Por-TT COF and COF-366-Zn were employed to illustrate the enhancement of metal catalytic sites and D–A structures for the photocatalytic performance.

## 1. Introduction

The pursuit of clean and sustainable energy sources is a desperate exploration to mitigate the consumption of fossil fuels and environmental pollution problems [1]. Hydrogen (H_2_) is a representative ideal green energy resource with a high energy density and free carbon emissions. A promising and sustainable path for hydrogen generation is photocatalytic water splitting under visible light, which is propelled by ubiquitous solar energy [2,3]. It has been demonstrated that photocatalysts play a decisive role in facilitating the kinetics of the H_2_ evaluation process. Consequently, the exploration of effective photocatalysts utilized to drive a H_2_ generation under visible light is of considerable significance [4].

Covalent organic frameworks (COFs) are an emerging kind of crystalline porous organic polymers, which possess properties such as a per-designed periodic structure, an inherent open channel, a large specific surface area, a satisfactory chemical/thermal stability, etc. [5,6,7,8,9,10,11]. These aforementioned characteristics enable COFs to function as potential desired photocatalysts for the generation of H_2_. However, the narrow absorption of visible light and the recombination of photon-generated charge carriers are key factors restricting the photocatalytic performance of COFs. The incorporation of donor–acceptor (D–A) combinations via a typical bottom-up design strategy is a reasonable strategy to broaden the absorption of visible light and promote the separation of photogenerated carriers, so that a series of D–A COFs have been designed and synthesized to function as photocatalysts for the evolution of H_2_ [12,13,14]. Among them, porphyrin, especially electron-deficient metalloporphyrin contained COFs, usually displayed a splendid catalytic performance thanks to the good visible light absorption ability and additional metal active sites [15,16,17,18,19,20,21]. Moreover, we noticed Zn-porphyrin-based COFs possess a longer-lasting state of the separation of electrons and holes than cobalt and nickel-coordinated porphyrin, which results from a suppressed ligand-to-metal charge transfer process caused by 3d^10^ configuration of the Zn^2+^ ion [22]. Therefore, Zn (II)-porphyrin-based COFs would be potential ideal photocatalysts for an efficient H_2_ evolution.

Herein, we select Zinc (II) 5,10,15,20-tetrakis(para-aminophenyl) porphyrin (Zn-TAPP) as the acceptor and thieno[3,2-b]thiophene-2,5-dicarbaldehyde (TT) as the donor to fabricate a metalloporphyrin-based D–A COF by the solvothermal method. The obtained Zn-Por-TT COF presented an excellent crystallinity, a large surface area, a good thermal/chemical stability, and a narrow band gap. As expected, the Zn-Por-TT COF demonstrated a good activity of photocatalytic hydrogen production and the hydrogen evolution rate of the Zn-Por-TT COF could attain to 8200 μmol/g/h in the presence of Pt co-catalyst and ascorbic acid sacrificial agent, which is above the average level among the reported COF photocatalysts. More importantly, a metal-free Por-TT COF and COF-366-Zn were synthesized and employed as counterparts to confirm that Zn^2+^ functions as the catalytic active site and the enhanced D–A interactions are beneficial for promoting the photocatalyst performance.

## 2. Results and Discussion

The Zn-Por-TT COF was synthesized in the mixed solvent system of *o*-dichlorobenzene/benzyl alcohol/6M AcOH at 120 °C for 3 days and what was obtained was a dark purple powder with a yield of 75% (Figure 1a). The successful co-condensation of Zn-TAPP and TT was unambiguously detected by Fourier transform infrared (FT-IR) spectroscopy (Figure 1b), the characteristic singles for -NH_2_ (~3352 and 3316 cm^−1^) and -CHO (~1660 cm^−1^) were dramatically decreased in Zn-Por-TT COF, and a new peak was able to be observed at ~1612 cm^−1^, which corresponded to the stretching vibration of C=N linkages. In addition, the single peak which appeared at ~150.4 ppm in the ^13^C solid-state NMR spectrum (Figure 1c) was able to ascribe to the carbon of the imine linkages, which further indicates the existence of C=N linkages. In addition, an X-ray photoelectron spectroscopy (XPS) measurement was carried out to evaluate the valence state of zinc in the prepared Zn-Por-TT COF (Figure 1d,e). The strong peaks which appeared at 1021.59 eV and 1044.60 eV should be attributed to Zn 2p_2/3_ and Zn 2p_1/2_, demonstrating that the valence state of Zn was +2. All these results indicate the efficient polymerization of Zn-TAPP and TT monomers.

The crystalline structure of the Zn-Por-TT COF was assessed by a powder X-ray diffraction measurement (PXRD) combined with the theoretical simulation (Figure 2a). There are at least six prominent diffraction peaks in the PXRD pattern, indicating the high crystallinity of Zn-Por-TT COF. The diffraction peaks appeared at 3.3°, 6.6°, 7.2°, 9.8°, 10.7°, and 22.2°, which can be assigned to (100), (200), (210), (300), (310), and (001) facts, respectively. The crystalline architecture of the Zn-Por-TT COF matched well with the simulated AA stacking model which provided a unit cell parameter of a = 27.23 Å, b = 27.34 Å, c = 4.29 Å, *α* = 76.3°, *β* = 100.9°, and *γ* = 95.8°. The calculated PXRD profile corresponds well to the experimental data and the deviations of the Pawley refinement are *R*_p_ = 2.76% and *R*_wp_ = 4.29%.

To explore the permanent porosity of Zn-Por-TT COF, the N_2_ sorption experiment was proceeded at 77 K and the samples were degassed at 150 °C for 8 h under a vacuum (10^−5^ bar) before their analysis (Figure 3). As shown in Figure 3a, the Zn-Por-TT COF revealed characteristic type-IV sorption isotherms, which possessed an obvious step at P/P_0_ = 0.05–0.18, demonstrating the existence of mesoporous. The Brunauer–Emmett–Teller (BET) surface area of the Zn-Por-TT COF was 632 m^2^/g, calculated from the absorption curve. The pore size distribution was analyzed by the nonlocal density functional theory (NLDFT) method and afforded an average pore size of 2.6 nm, which is approximated with the theoretical value (2.5 nm). Moreover, the morphology feature of the Zn-Por-TT COF was assessed by scanning electron microscopy (SEM) and transmission electron microscopy (TEM). As revealed by the SEM images (Figure 4a,b and Appendix A), the Zn-Por-TT COF presented random nanoflakes with an average size of ~200 nm. Additionally, clear lattice textures with an interval of 2.6 nm were readily observed in the TEM images, which attests the high crystalline feature and long-rang ordered structure of the Zn-Por-TT COF (Figure 4c,d and Appendix A). In addition, the energy-dispersive X-ray spectroscopy (EDS) elemental mappings revealed that C, N, S, and Zn were uniformly distributed in these nanoflakes (Figure 4e–h).

Additionally, the Zn-Por-TT COF showcased an excellent thermal stability which was confirmed by the thermogravimetric analysis (TGA) measurement under N_2_. The distinct weight loss appeared at ca. 500 °C, indicating that the skeleton of the Zn-Por-TT COF started to collapse (Figure 5a). The chemical stability of the Zn-Por-TT COF was detected by immersing the COF powder into the solution of hydrochloric acid (pH = 2) and 1 M of sodium hydroxide solution for 24 h. After these treatments, the crystallinity of the Zn-Por-TT COF remained well, indicating a good chemical stability of the Zn-Por-TT COF (Figure 5b). Furthermore, the UV-vis diffuse reflectance spectroscopy (UV-DRS) revealed that the Zn-Por-TT COF exhibited a broad visible light absorption capacity in the range of 250~700 nm and resulted in a narrow optical band gap of ~1.8 eV (Figure 6a). The Mott–Schottky measurement was further performed to determine the conduction band (CB) of Zn-Por-TT COF and it provided a value of −1.43 V vs. Ag/AgCl (−1.23 V vs. NHE) (Figure 6b). Subsequently, the corresponding valence band (VB) was estimated to be 0.57 V vs. NHE [23]. The band structure satisfies the photocatalytic water splitting property in thermodynamics.

A good chemical stability, an excellent visible light absorption capacity, an inherent porosity, and a suitable bond gap render the Zn-Por-TT COF an alternative photocatalyst for a H_2_ evolution under a visible light illumination. To demonstrate this hypothesis, the photocatalytic hydrogen production experiments were performed to evaluate the photocatalytic performance of a Zn-Por-TT COF, and the general H_2_ evaluation procedure is described as below. Primarily, 5 mg of the Zn-Por-TT COF powder was dispersed in 50 mL of ultrapure water, a few amounts of ascorbic acid (AA), and K_2_PtCl_6_ were added into the system which functions as an electron sacrificial agent and co-catalyst to produce hydrogen, respectively. The visible light (λ > 400 nm, xenon lamp) was employed as the optical source and the photocatalytic system should be kept at 4 °C during the process of a H_2_ evolution. In order to identify the optimal conditions for a H_2_ evolution catalyzed by Zn-Por-TT COF, the effects of the pH and co-catalyst ratio were investigated unambiguously. The optimal pH value was revealed to be three (Figure 6d) and the most appropriate amount of Pt co-catalyst was 5 wt% (Figure 6e), which both displayed higher H_2_ evolution rates compared with the control experiments. Intriguingly, the Zn-Por-TT COF displayed a moderate H_2_ evolution rate of 4100 μmol/g/h at the first 5 h while the rate could enhance steadily to attain 8200 μmol/g/h after the illumination for 10 h and was maintained for at least 40 h (Figure 6f), which is comparable and better than many other reported COF materials (Table 1). The enhancement of the hydrogen production rate in first 10 h could be attributed to the progressive deposition of Pt [24,25,26]. In addition, the chemical constitution, morphology, and crystalline structure of the Zn-Por-TT COF were able to remain well after photocatalysis, and the elements of C, N, S, Zn, and Pt were uniformly distributed or deposited in the Zn-Por-TT COF nanoparticle (Figure 7), which demonstrates the splendid cycle stability of Zn-Por-TT COF.

To investigate the enhancement effect of the D–A structure and Zn (II) coordination center on the photocatalytic performance, COF-366-Zn without the TT donor units and a Por-TT COF free of Zn (II) coordination center were constructed and employed to function as contrasted photocatalysts for the H_2_ evolution. The chemical constitution, crystallinity, and morphology of COF-366-Zn and Por-TT COF were unequivocally characterized by PXRD, FT-IR, XPS, etc. (Figure 8a–f and Appendix A). Both COF-366-Zn and Por-TT COF exhibited a similar broad absorption from approximately 250 to 700 nm and narrow band gaps which presumably caused by the incorporation of porphyrin (Figure 9a). However, the H_2_ evolution rates of the two contrasted COFs were significantly reduced (Figure 9b) and the photocurrents of COF-366-Zn and Por-TT COF were weaker than the Zn-Por-TT COF (Figure 9c), which suggests the faster charge transfer capacity of the Zn-Por-TT COF and indicates the D–A structure and the Zn (II) coordination center synergistically promoted the photocatalytic performance of Zn-Por-TT COF.

## 3. Conclusions

In summary, a Zn(II)-porphyrin built-in COF was successfully fabricated via a reasonable design and the obtained Zn-Por-TT COF exhibited a high crystallinity, a good stability, and a broad absorption capacity of visible light so that can be employed as an effective photocatalyst for a H_2_ generation. As our hypothesis, the Zn-Por-TT COF exhibits a satisfactory H_2_ evolution rate which can be stabilized at ~8200 μmol/g/h eventually. Moreover, the formed D–A structures and abundant metal catalytic active sites synergistically promote the H_2_ production performance of the Zn-Por-TT COF, as demonstrated by the controlled experiments. This work provides an available path for the construction of highly efficient photosensitizers for solar energy harvesting and conversion. We envision that COF photocatalysts with desirable properties will be constructed via incorporating stronger D–A complex structures, and related works are in progress in our laboratory.

## Figures and Tables

**Figure 1 polymers-14-04893-f001:**
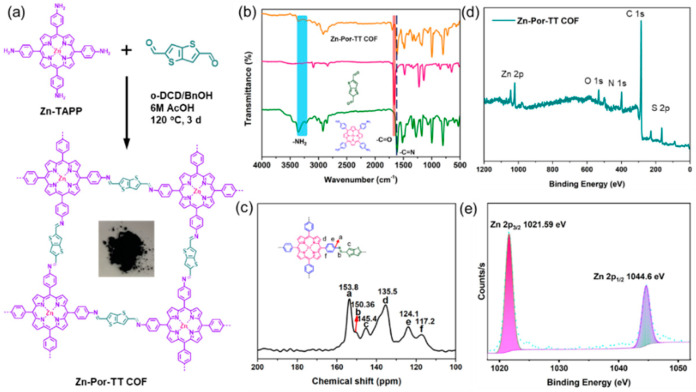
(**a**) Synthetic diagram of Zn-Por-TT COF; (**b**) FT-IR spectra of Zn-Por-TT COF (orange), TT (pink), Zn-TAPP (green); (**c**) ^13^C CP-MAS NMR spectrum of Zn-Por-TT COF; (**d**) XPS of Zn-Por-TT-COF; and (**e**) high-resolution XPS spectrum of Zn.

**Figure 2 polymers-14-04893-f002:**
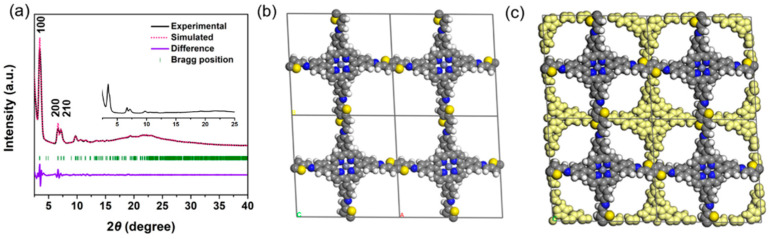
(**a**) Experimental PXRD patterns of Zn-Por-TT COF (black), Pawley refinement (pink), their difference (violet), and Bragg positions (green); (**b**) AA-stacking; and (**c**) AB-stacking models of Zn-Por-TT COF.

**Figure 3 polymers-14-04893-f003:**
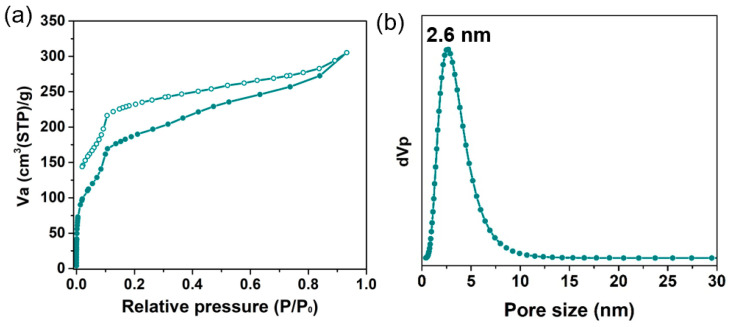
(**a**) N_2_ absorption isotherms of Zn-Por-TT COF at 77 K; (**b**) pore size distribution profile of Zn-Por-TT COF.

**Figure 4 polymers-14-04893-f004:**
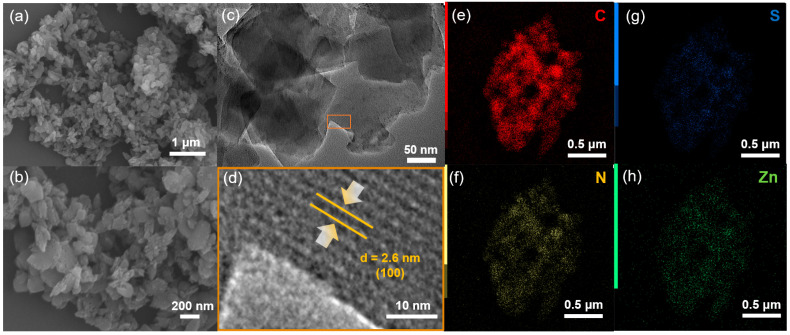
(**a**,**b**) SEM and (**c**,**d**) TEM images of Zn-Por-TT COF; (**e**–**h**) elemental maps of Zn-Por-TT COF.

**Figure 5 polymers-14-04893-f005:**
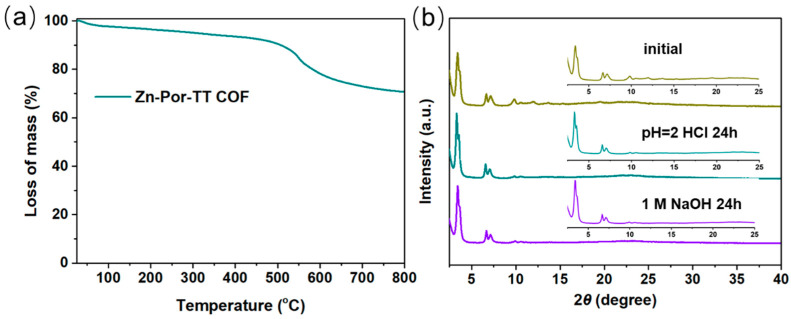
(**a**) TGA profile of Zn-Por-TT COF in N_2_; (**b**) PXRD patterns of Zn-Por-TT COFs before (dark yellow) and after treatments in the solution of HCl (pH = 2, cyan) and NaOH (1 M, violet).

**Figure 6 polymers-14-04893-f006:**
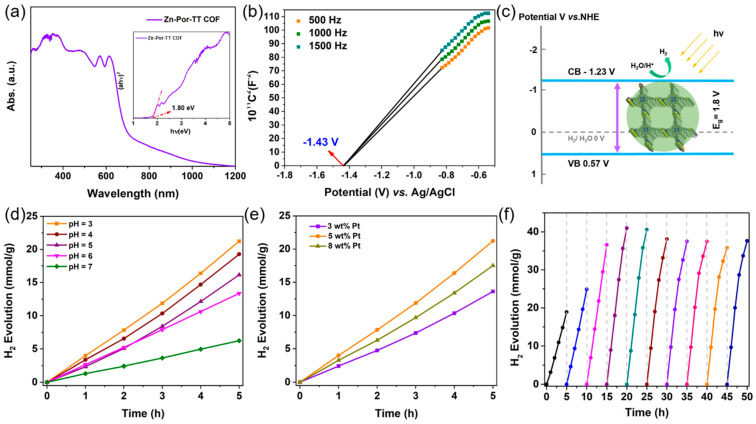
(**a**) UV-Vis DRS spectrum of Zn-Por-TT COF, insert: (ahυ)^2^ versus hυ curve of Zn-Por-TT COF; (**b**) Mott–Schottky (M-S) plots of Zn-Por-TT COF at 500 Hz, 1000 Hz, 1500 Hz; (**c**) the CB and VB positions of Zn-Por-TT COF; (**d**) H_2_ evolution rate of Zn-Por-TT COF under different acidity; (**e**) H_2_ evolution rate of Zn-Por-TT COF with different Pt loading amounts; (**f**) time course of H_2_ productions under visible light (λ > 400 nm) irradiation of Zn-Por-TT COF.

**Figure 7 polymers-14-04893-f007:**
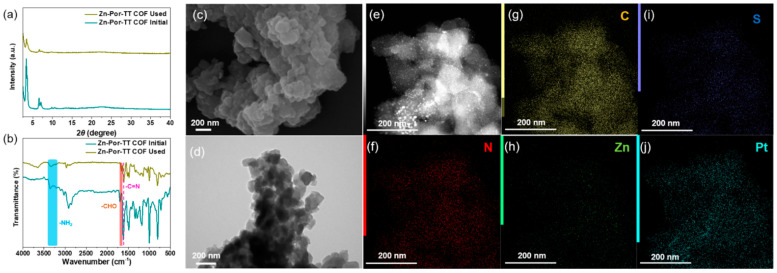
(**a**) PXRD pattern of Zn-Por-TT COF before and after photocatalysis; (**b**) FT-IR spectra of Zn-Por-TT COF before and after photocatalysis; (**c**) SEM of Zn-Por-TT COF after photocatalysis; (**d**) TEM of Zn-Por-TT COF after photocatalysis; and (**e**–**j**) EDS Maps of Zn-Por-TT COF after photocatalysis.

**Figure 8 polymers-14-04893-f008:**
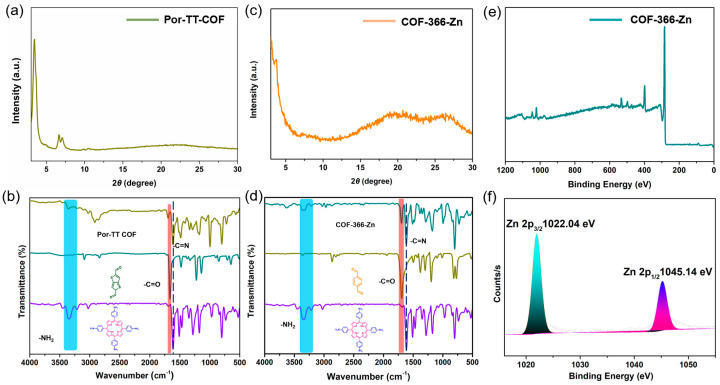
(**a**) PXRD pattern of Por-TT COF; (**b**) FT-IR spectra of Por-TT-COF (dark yellow), TT (cyan), TAPP (violet); (**c**) PXRD pattern of COF-366-Zn; (**d**) FT-IR spectra of COF-366-Zn (cyan), BDT (dark yellow), TAPP (violet); (**e**) XPS of COF-366-Zn; and (**f**) high-resolution XPS spectra of Zn.

**Figure 9 polymers-14-04893-f009:**
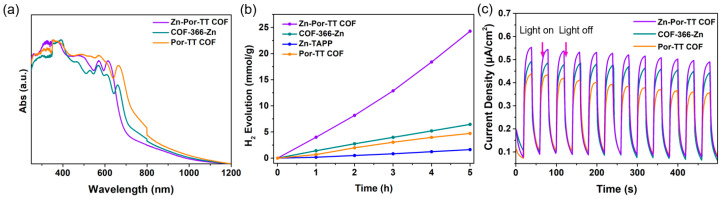
(**a**) DRS-UV spectra of Zn-Por-TT COF (violet), COF-366-Zn (cyan), and Por-TT COF (orange); (**b**) H_2_ evolution rates of Zn-Por-TT COF (violet), COF-366-Zn (cyan), and Por-TT COF (orange); (**c**) photocurrent-time plots of Zn-Por-TT COF (violet), COF-366-Zn (cyan), and Por-TT COF (orange).

**Table 1 polymers-14-04893-t001:** The photocatalytic performance comparison of Zn-Por-TT COF with other representative COF based photocatalysts.

Name	Activity (μmol/g/h)	Conditions	Ref.
Ni-Py-COF	13,231	Pt (8 wt%), 5 mg AA in 10 mL water, 800 nm > λ > 380 nm	[20]
Ni-Bn-COF	1805		
PETZ-COF	7204.3	Pt (3 wt%), 0.1 M AA, λ > 420 nm	[13]
PEBP-COF	217.1		
BTH-1	10,500	Pt (8 wt%), 0.1 M AA, λ > 420 nm	[27]
BTH-2	1200		
BTH-3	15,100		
COFs-1	1033	Pt (1 wt%), TEOA (10 vol %), λ > 420 nm	[28]
COFs-2	1444		
COFs-3	2789		
COFs-4	1274		
Py-CITP-BT-COF	8875	Pt (5 wt%), 0.1 M AA, λ > 420 nm	[29]
Py-FTP-BT-COF	2875		
Py-HTP-BT-COF	1078		
RC-COF-1	27,980	Pt (3 wt%), 0.1 M AA, λ > 420 nm	[30]
Zn-Por-TT COF	8200	Pt (5 wt%), 0.1 M AA, λ > 400 nm	This work

## Data Availability

The data presented in this study are available on request from the corresponding author.

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
