# Peer review of "Zn (II) Porphyrin Built-in D–A Covalent Organic Framework for Efficient Photocatalytic H2 Evolution"

_polymers, 2022, doi:10.3390/polym14224893_

Round 1

Reviewer 1 Report

The manuscript can be reconsidered after revision. Below are comments and suggestions to help the authors improve it:

1.Figure 1 d assign the all peaks in xps survey. after deconvolution

 assign other relevant peaks.

2. Figure 2a and Figure 5b , need low angle XRD  for clearly shown all peaks between 0 to 25o

3.In TGA graph Mass% in Y axis in place of weight%.

4. In FTIR spectra assign the major peaks(Figure 7b)

5. In conclusion section: add the future prospects 2-3 lines.

Author Response

Reply to the Comments of Reviewer 1

Reviewer: 1

The manuscript can be reconsidered after revision. Below are comments and suggestions to help the authors improve it:

Reply: We thank the honorable reviewer for appreciating the positive comments and valuable suggestions on our work. We had tried our best to modify our manuscript according to the issues raised by the reviewer.

  1. Figure 1 d assign the all peaks in xps survey after deconvolution assign other relevant peaks.

Reply: We appreciate the reviewer’s kind comments. Accordingly, we had assigned all the peaks reasonably in Figure 1d.

  1. Figure 2a and Figure 5b , need low angle XRD for clearly shown all peaks between 0 to 25o.

Reply: We appreciate the reviewer’s kind suggestions. We had provided insert PXRD patterns range from 2-25o for Figure 2a and Figure 5b to clearly show all diffraction peaks in our revised manuscript.

  1. In TGA graph in Y axis in place of weight%.

Reply: We appreciate the reviewer’s kind suggestions. We had changed weight% as Mass% in Y axis of the TGA graph.

  1. In FTIR spectra assign the major peaks (Figure 7b).

Reply: We appreciate the reviewer’s kind comments. We had assigned the major peaks of IR spectra in Figure 7b.

  1. In conclusion section: add the future prospects 2-3 lines.

Reply: We appreciate the reviewer’s kind comments. As suggested by the honorable reviewer, we have added a short sentence of “We envision that COF photocatalysts with desirable properties will be constructed via incorporating stronger D-A complex structures, and related works are in progress in our laboratory.” in conclusion part as the prospect to make our conclusion more comprehensive.

Reviewer 2 Report

The manuscript by Lv et. al. reported fabrication of imine-linked Zn-Por-TT covalent organic frameworks for photocatalytic hydrogen evolution. Furthemore, the phase purity, chemical stability, thermal stability, porosity is well investigated. This work is of interest to readers and could be suitable for Polymers, MDPI after consideration of the following points:
1. The authors should explain the activation condition, degas condition of COF. Furthemore, desorption isotherm (Figure 3) of COF should be revised. 
2. On Page 6, Figure 7, the used Zn-Por-TT COF PXRD exhibited slight change in PXRD. The plausible reason should be explain, furthermore, surface area analysis should be performed.
3. The authors should cite Chem. Rev. 2020, 120, 16, 8814-8933, Nanoscale, 2019, 11, 21679-21708. 

4. The authors should revised, line 76 on Page 2. It should be "the single peak appeared at..."

Round 2

Reviewer 1 Report

Dear  Author,

This manuscript can be accepted as present form

Reviewer 2 Report

The manuscript by Lv et. al. reported fabrication of imine-linked Zn-Por-TT covalent organic frameworks for photocatalytic hydrogen evolution. Furthemore, the phase purity, chemical stability, thermal stability, porosity is well investigated. The manuscript have been carefully revised, addressed all the comments/suggestions proposed by the reviewers and is now suitable for polymers, MDPI.